# Metabolic and Antioxidant Variations in “Regina” Raspberries: A Comparative Analysis of Early and Late Harvests

**DOI:** 10.3390/plants14060888

**Published:** 2025-03-12

**Authors:** María Teresa Sanchez-Ballesta, Claudia Balderas, María Isabel Escribano, Carmen Merodio, Irene Romero

**Affiliations:** Department of Characterization, Quality and Safety, Institute of Food Science, Technology and Nutrition (ICTAN-CSIC), Jose Antonio Novais 6, 28040 Madrid, Spain; mballesta@ictan.csic.es (M.T.S.-B.);

**Keywords:** *Rubus idaeus*, harvest periods, polyphenols, antioxidant capacity, metabolic profiling

## Abstract

Raspberries (*Rubus idaeus* L.) are valued as both a food source and a medicinal plant, with expanding production driven by premium market demand. Primocane varieties, such as “Regina” are prized for their ability to produce two harvests per year, offering extended availability into autumn and providing significant commercial benefit. Their high polyphenol content, particularly in anthocyanins and flavonoids, contributes to antioxidant and health properties. However, their extraction and quantification are influenced by factors such as genetic variability, environmental conditions, fruit maturity, storage, and processing techniques. This study analyzed the metabolic profiles of “Regina” raspberries harvested in June (RiJ) and September (RiS). Out of 748 compounds listed in the Phenol-Explorer database, 377 metabolites were identified and categorized. Principal component analysis (PCA) revealed significant differences between harvests, with RiS samples showing higher concentrations of most flavonoid and non-flavonoid compounds. Heatmap and volcano plot analyses confirmed that significant metabolites were enriched in RiS samples. Correspondingly, antioxidant capacity, measured via ABTS and FRAP assays, was higher in RiS raspberries. These findings highlight the metabolic and antioxidant differences between harvest periods and lay the groundwork for understanding how these compounds could be modulated through the application of postharvest treatments.

## 1. Introduction

Raspberries are a highly valued fruit, both as a food source and for their medicinal properties [1]. Historically, most raspberry production has been sold to industries for commercialization as frozen fruit, for jam production and canning, or for use in the making of juices and flavorings for ice cream, yogurt, and other products. However, in recent years, production for the fresh market has grown significantly and has become a key segment of the industry [2]. In 2023, global raspberry production averaged 940,979.29 tons. Among berry crops, raspberries rank second in yield growth rate, surpassed only by blueberries. In 2023, Russia was the world’s leading producer of raspberries, followed by Mexico, Serbia, Poland, and the United States, which together ranked as the top five producers globally [3].

The genus *Rubus* includes numerous described species, of which only about 331 are officially recognized. Among them, *Rubus idaeus* var. *idaeus*, commonly known as the red raspberry or European red raspberry, is the most notable in Europe. The genotypes commonly known as primocane raspberries, including American varieties (“Heritage”, “Summit”, and “Autumn bliss”) and European varieties (“Polka”, “Sugana”, and “Regina”), among others, produce an early summer harvest on canes from the previous year, followed by another crop from late summer to early autumn on canes grown during the current year. They yield an average of 15 to 20 t/ha, with the peak production occurring during the second harvest. These varieties are noteworthy from a postharvest perspective because their fruit remains available into the autumn, which represents a commercial advantage.

Raspberry fruits have been used for therapeutic purposes for a long time [4,5]. They provide numerous health benefits, including antioxidant, anti-aging, anticancer, lipid-lowering, and weight-loss properties, making them a promising candidate for reducing the risk of chronic diseases and fostering the development of therapeutic foods [5,6]. In *Rubus idaeus*, a total of 50 polyphenols were identified and quantified across 4 different forms of red raspberries [7], whereas, in *Rubus chingii* Hu raspberries, 105 distinct compounds have been identified [4], including major components such as phenols, flavonoids, terpenes, organic acids, alkaloids, phenylpropanoids, coumarins, glycoproteins, flavonoid glycosides, polysaccharides [8], and vitamins (A, B1, B2, B5, B6, C, and E). Furthermore, the unique aggregate structure of raspberries enhances their nutritional profile by increasing their dietary fiber content, which is among the highest found in whole foods, with fiber accounting for to 6% of their total weight [9]. This diverse compound profile, with phenols predominating in both quantity and antioxidant relevance, contributes to the nutritional value of raspberries and their health benefits.

In most berries, it is known that the levels of flavonoids and anthocyanins increase as the fruit ripens. However, the evaluation of total phenolic and total anthocyanin content in *R. idaeus* cultivars (“Reveille”, “Tulameen”, and “Heritage”) across five developmental stages revealed that unripe raspberries contained high levels of non-anthocyanin phenols [10]. Consequently, the precise quantification of phenolic phytochemicals can exhibit considerable variability, not only due to differences in extraction and measurement techniques but also due to genetic diversity among cultivars and the influence of environmental factors such as ripeness, cultivation conditions, storage, and processing methods [11,12].

The “Regina” raspberry cultivar was developed in Italy in 2009. It is known for its high hardiness and suitability for fresh consumption. Being a primocane variety, it is particularly valued for its yield and the quality of its fruit [13]. Despite the increasing interest in raspberry fruit, there is limited knowledge about its metabolomic profile at the time of harvest when it is ready for consumption. To the best of our knowledge, this study is the first to analyze the phenolic composition of “Regina” raspberries harvested in two different seasons using HPLC/QTOF-MS. The aim of this work is to analyze the metabolic profiles of “Regina” raspberries harvested at two different times (June and September) to investigate the differences in metabolic compounds and antioxidant capacity between the harvests. The results show that the RiS samples accumulated the most metabolites and exhibited the highest antioxidant capacity, particularly in ABTS measurements. This study provides a starting point for future research investigating how these fruits from different harvest periods, each with distinct metabolic profiles, respond to various postharvest treatments and how these compounds can be modulated to maintain raspberry quality.

## 2. Results and Discussion

### 2.1. Phenolic Composition Analysis

To gain a comprehensive understanding of the phenolic composition of “Regina” raspberries from two harvest periods, June (RiJ) and September (RiS), targeted metabolomics was performed using HPLC/QTOF-MS in both positive and negative ionization modes. Starting from a database containing information on 748 compounds according to the Phenol-Explorer database, a total of 739 (positive mode) and 390 (negative mode) molecular entities, or “features”, were initially generated. Each of these signals is considered a “fingerprint”. The integration performed using the software was evaluated for accuracy, and when it was found to be incorrect, a manual integration of the signals was carried out by applying a series of filters. Minor peaks without available standards or typical spectra were not identified. Figure 1 shows two examples of representative polyphenol groups in raspberries, as listed in the Phenol-Explorer: an anthocyanin (cyanidin-3-glucoside) from the flavonoid group and a hydroxybenzoic acid (gallic acid) from the non-flavonoid group. The fragmentation patterns of some of the most representative compounds in raspberries, according to the Phenol-Explorer, are listed in Appendix A. This process resulted in a final total of 377 molecular entities, of which 207 corresponded to the positive mode and 170 to the negative mode.

### 2.2. Comparative Analysis of Flavonoids and Non-Flavonoids Compounds

Data obtained from both ionization modes were analyzed using PCA to investigate the sample distribution (Figure 2). The PCA effectively distinguished between RiJ and RiS samples, with principal component 1 explaining 99% of the variance in the positive mode (Figure 2A) and 96% in the negative mode (Figure 2B), indicating significant metabolic differences between the two harvest periods. The influence of growing conditions and harvest times on the phenolic composition of raspberries has been previously highlighted by [14], who examined the quality and chemical composition of ten red raspberry genotypes over three harvesting seasons, though the cultivar “Regina” was not part of their study. They found significant differences in terms of major phenolic compounds (cyanidin derivatives, anthocyanins, tannins, ellagic acid derivatives, quercetin derivatives, and the sum of all detected phenolic compounds) among cultivars and across years of harvesting, with the condensed tannin Lamber-tianin C being the only metabolite stable across years. These results agree with ours, where harvesting time emerges as a significant variable that clearly separates RiJ and RiS samples.

After duplicated compounds were eliminated from both ionization modes, 224 unique metabolites were identified. These were classified as follows: flavonoids (65.6%), phenolic acids (25%), lignans (3.6%), stilbenes (0.4%), and other phenolic compounds (5.4%) (Figure 3).

The average composition of metabolites per harvest month was analyzed with respect to the two primary groups of polyphenols: flavonoids and non-flavonoids. Within the flavonoid group (Figure 4A), the concentrations of anthocyanins, dihydrochalcones, flavones, and isoflavonoids were significantly higher in RiS samples, approximately twice those found in RiJ. Although flavanones and flavonols were also more abundant in RiS, flavanol levels were comparable between the two groups. The higher flavonoid concentrations observed in this study are not consistent with the findings from [14], where middle-early harvested raspberries, such as “Veten” and “RU98406038”, exhibited high flavonoid concentrations. Thus, ref. [15] emphasizes that seasonal and ripening factors can lead to increases in secondary metabolites in raspberries, including anthocyanins and phenolic acids, which contribute to their antioxidant profile. When focusing on the non-flavonoid metabolites (Figure 4B), it is evident that hydroxybenzoic acids, such as gallic acid, and hydroxyphenylpropanoic acids, including p-coumaric acid, are highly abundant, with concentrations in the RiS samples approximately twice those found in RiJ. However, the high variability observed in these compounds across the samples makes the differences between groups not statistically significant. Other compounds, including hydroxycinnamic acids, hydroxyphenylacetic acids, various phenolic compounds, lignans, stilbenes, and furanocoumarins, were also more abundant in RiS, with the last three being the only ones that showed significant differences between RiJ and RiS. In contrast, hydroxybenzaldehydes, hydroxycoumarins and tyrosols were significantly more abundant in the RiJ samples, which may be related to a defense mechanism against different biotic or abiotic stresses that could be more prevalent during that harvest period, although this remains a hypothesis. Overall, the RiS samples exhibited a higher average composition of both flavonoid and non-flavonoid compounds. This is consistent with the results from [14], which showed that late-harvested raspberry genotypes had higher levels of total phenolic compounds.

The significant contribution of both groups, particularly anthocyanins from flavonoid metabolites and ellagic acid derivatives along with other phenolic acids from non-flavonoid metabolites, is well established in enhancing antioxidant activity and supporting the health-promoting properties of these fruits [16].

### 2.3. Heatmap and Volcano Plot Analysis

The heatmap of the flavonoid group showed a high presence of anthocyanins, flavonols, flavones, and chalcones in both groups of raspberries, although higher concentrations were found in RiS samples (Figure 5A). Among the analyzed anthocyanins, cyanidin, peonidin, and pelargonidin derivatives were found to be more abundant in the RiS samples, with cyanidin 3-sophoroside being the most prevalent anthocyanin in both RiJ and RiS. These results are consistent with [7], who reported that cyanidin-, pelargonidin-, and peonidin-based molecules were the predominant antocyanins found in red raspberries. Furthermore, [17] noted that cyanidin 3-sophoroside can serve as a distinguishing marker for cultivated red *Rubus* species, as it is typically the major anthocyanin present in these varieties. Additionally, among the flavonols, kaempferol derivatives were slightly more abundant in RiS samples. Isoflavonoids also showed higher concentrations in RiS samples, whereas quercetin glucosides were more abundant in RiJ samples. Similarly, catechin, epicatechin, and epicatechin–glucuronides were present in greater amounts in the RiJ samples.

The heatmap for non-flavonoids polyphenols (Figure 5B) showed a greater amount of these compounds in the RiS samples, with the most abundant being hydroxybenzoic acids (syringic acid, ethyl gallate, and 3,4-O-dimethylgallic acid), hydroxycinnamic acids (feruoylquinic acid derivatives), and a hydroxyphenylpropanoic acid (3-(3,4-dihydroxyphenyl) lactic acid). However, a group of hydroxybenzoic acid derivatives (4-hydroxybenzoic acid 4-O-glucoside, ellagic acid arabinoside, 3,5-dihydroxy-4-methoxybenzoic acid, and their respective isomers), as well as tyrosols, were more abundant in the RiJ samples. The role of these compounds is well known for inducing a range of defense mechanisms that enhance resistance to various diseases. Salicylic acid, a derivative of hydroxybenzoic acid, has been shown to activate an immune response known as “systemic acquired resistance”, which provides broad-spectrum defense against pathogens [18]. Furthermore, tyrosols possess antioxidant and antimicrobial properties, which contribute to plant defense by protecting them from oxidative stress and pathogenic attack [19]. Notably, lignans such as sesamolin and sesaminol derivatives were present in the RiS samples. These bioactive compounds are found in various plants, including sesame seeds (*Sesamum indicum* L.), although their presence and function in raspberries have not been thoroughly studied. Besides their effects on plants, some of these lignans, such as sesamin and sesamolin, possess beneficial properties for humans, as they have demonstrated antioxidant, anti-inflammatory, and antimicrobial activities [20].

The combination of heatmap and volcano plot visualization methodologies facilitated the identification and illustration of statistically significant variations in metabolite abundance between the two flowering periods under investigation, RiJ and RiS (Figure 6). Using a fold-change threshold of 2 and a *t*-test cut-off of 0.05, significant metabolites were highlighted. In the plot, features located in the upper-right corner represent those with significant changes and higher concentrations in RiJ samples. Similarly, features located in the upper-left corner represent those with significant changes and lower concentrations in RiJ samples.

Thus, on the right upper corner, the 4-hydroxybenzoic acid 4-O-glucoside, poncirin, and isorhamnetin glucuronide derivatives are highlighted, as are catechin, epicatechin, esculin, and cyanidin derivatives, among others. On the upper-left corner, quercetin glucuronide derivatives, delphinidin 3-glucoside, and myricetin derivatives appear to be less present in RiJ samples. Delving deeper into the topic, Appendix A lists all the significant metabolites between the two conditions, RiJ and RiS samples, derived from the volcano plot analysis. Key metrics include fold change (FC), its logarithmic transformation (log_2_(FC)), adjusted *p*-values (p.adjusted), and statistical significance expressed as −log_10_(*p*-value). The most significant compounds, such as apigenin 6,8-C-arabinoside-C-glucoside and apigenin 6,8-C-galactoside-C-arabinoside, exhibit high −log_10_(*p*-values~4.5575) and notable negative log_2_(FC) values (~−2.3), indicating a strong decrease in relative abundance. Similarly, compounds like procyanidin trimer EEC and procyanidin trimer C2 show a significant decrease in abundance (log_2_(FC)~−2.1) with strong statistical support (*p*.adjusted~3.56 × 10^−5^). Interestingly, among the top 60 most significant metabolites, only one feature, 4-hydroxybenzoic acid 4-O-glucoside, exhibits positive fold changes (log_2_(FC) = 1.978), indicating an increase in relative abundance in the RiJ samples. Other compounds that were significantly more abundant in the RiJ samples were flavonoid glucosides. It has been reported that the concentration of hydroxybenzoic acids in fruits and vegetables is typically low, and it is predominantly found in conjugated forms, with levels in raspberries ranging between 32 and 56 mg/kg FW [21]. The feature present in higher concentrations in RiJ samples consists of a 4-hydroxybenzoic acid conjugated with a glucose molecule through a glycosidic bond at the 4-hydroxy position, and it belongs to the group of phenolic glycosides.

Glycosylation plays a key role in increasing the solubility and stability of phenolic acids, enhancing their transport and storage in plant tissues. In plants, compounds like 4-hydroxybenzoic acid 4-O-glucoside can be involved in defense mechanisms against pathogens and oxidative stress, among others [22]. Following the trend of the results observed in this work, most of the metabolites with higher values of −log_10_(*p*-values) were highly abundant in RiS samples (Appendix A). Flavonoids, such as apigenin derivatives, and some flavones such as orobol, luteolin, and scutellarein, among others, dominate the list, underscoring their biological relevance. Additionally, procyanidins and anthocyanins, including cyanidin derivatives, as well as flavonoids like kaempferol derivatives, were also prominently represented. Similar results were observed by [23] when comparing raspberries from two different plateaus; the most significantly different metabolites were flavonoids and phenolic acids. These compounds are well known in plants for their antioxidant and anti-inflammatory properties, as well as their role in helping plants against abiotic stresses (e.g., UV radiation, drought, and temperature extremes) and biotic stresses (e.g., pests and pathogens) [24,25]. In raspberries, flavonoids and phenolic acids also hold commercial significance due to their impact on quality, appearance, and antioxidant properties, which are highly valued in the food and health industries [26].

### 2.4. Antioxidant Capacity Analysis

Given the observed higher concentrations of flavonoid and non-flavonoid metabolites in RiS samples, along with the predominance of significant compounds in these samples, it is reasonable to expect that they would exhibit greater antioxidant capacity. To test this, antioxidant capacity was measured using two methods with the single electron transfer mechanism: ABTS and FRAP assays [27]. As shown in Figure 7, the RiS samples demonstrated higher antioxidant capacity values using both methods (ABTS: 90.37 ± 7.87 µmol TE/gFW; FRAP: 65.99 ± 3.31 µmol TE/gFW) compared to the RiJ samples (ABTS: 71.81 ± 5.15 µmol TE/gFW; FRAP: 53.76 ± 5.39 µmol TE/gFW). Notably, the ABTS values were higher than those obtained using the FRAP method.

The ABTS method is widely used to assess the antioxidant capacity, particularly for polyphenols, due to its ability to neutralize both hydrophilic and lipophilic free radicals. In contrast, the FRAP method is a quick and straightforward technique that measures the reducing capacity of antioxidants in the sample by converting Fe^3+^ to Fe^2+^ under acidic conditions. Both methods are considered complementary tools for measuring total antioxidant capacity, with ABTS suited for radical-neutralizing capacity and FRAP for evaluating the specific reducing power of phenolic compounds in raspberries and other fruits [26,28].

However, it is important to note that kaempferol glycosides often show a minimal response in FRAP assays, whereas quercetin glycosides typically exhibit higher antioxidant activity, making them more readily detected through this method. The ABTS assay, on the other hand, can detect both kaempferol and quercetin derivatives, offering a broader measure of antioxidant capacity [29]. This aligns with our results, in which ABTS values were higher than FRAP values, as both kaempferol and quercetin glycosides were detected. Similar results (74.25 ± 1.47 μmol TE/g FW) were observed by [30] when the antioxidant capacity was measured via ABTS using a methanol extract from “Heritage” raspberries, another primocane cultivar.

Furthermore, it is important to highlight that the maturity index (MI) [soluble solid content (SSC)/titratable acidity (TA)] was higher in fruit harvested in September (MI RiS 4.06; MI RiJ: 3.35) due to slightly higher sugar content. It is known that polyphenols tend to decrease as the fruit matures due to their role in protecting against oxidative stress during the early stages of development [31], which is expected to be related to a decrease in antioxidant capacity. However, the results of this study indicated an increase in phenolic compounds and antioxidant capacity in RiS samples, despite their higher MI.

## 3. Material and Methods

### 3.1. Plant Material

The plant material used in this work consisted of primocane raspberries (*Rubus idaeus* L.) of the “Regina” cultivar obtained from two flowering periods [June (SSC: 8.00; TA: 2.39; MI: 3.35) and September (SSC: 10.80; TA: 2.66; MI: 4.06)], harvested in 2022 in Salas (Asturias), Spain. Uniform, disease-free fruit at commercial ripeness was randomly hand-harvested. The raspberries were collected in 250 g recycled polyethylene (rPET) trays with regular atmosphere (Infia K10 W/40 mm, Infia Ibérica, Valencia, Spain), each including a pad and a lid. The fruits were transported on the same day of harvest to the Institute of Food Science and Technology and Nutrition (ICTAN-CSIC, Madrid, Spain). Some fruits were analyzed for quality assessments, while the rest were frozen in liquid nitrogen, ground into a fine powder, and stored at −80 °C until analyses were performed.

### 3.2. Extraction Procedure

The extracts were obtained from 0.2 g of pulverized fruits stored at −80 °C, which were homogenized with 1 mL of a methanol–water mixture (50:50) acidified with 1% HCl. The mixture was incubated with agitation for one hour at room temperature. The samples were then centrifuged at 10,000× *g* for 10 min at room temperature, and the supernatant was collected. This process was repeated using the precipitate, and the supernatants were combined in the same Eppendorf tube to a final volume of 2 mL. The supernatants were filtered through 0.45 µm nylon filters, and the extracts were stored at −20 °C. Extractions were performed in three biological replicates for each sample.

### 3.3. Instrumentation and Experimental Conditions

The LC-QTOF-MS analysis was performed on an Agilent 1200-series liquid chromatography system binary pump coupled with an Agilent 6530 Accurate Mass Quadrupole Time-Of-Flight (QTOF) Mass Spectrometer (Agilent Technologies, Waldbronn, Germany) equipped with two binary pumps, an autosampler, a degasser, and a column heater.

Chromatographic separation was achieved on a Zorbax Eclipse XDB-C18 column (150 mm × 4.6 mm i.d., 5 μm). The gradient consisted of mobile phase A (0.1% formic acid (FA) in water) and B (0.1% FA in acetonitrile (ACN)). It started at 5% phase B and increased to 50% over 35 min, followed by a 2-min re-equilibration to initial conditions, and held for 7 min, bringing the total run time to 45 min. The sample injection volume was 5 µL, and the flow rate was maintained at 1 mL/min.

Mass spectrometry detection was performed in both positive and negative electrospray ionization modes, with a full scan range of 100 to 1100 *m*/*z*. The capillary voltage was set to 4000 V for both ESI (+) and ESI (−). Gas flow was maintained at 10 L/min at 325 °C with a nebulizer pressure of 45 psi. The fragmentor voltage was 125 V, and the skimmer and octupole radio frequency voltage (OCT RF Vpp) was set to 65 V and 750 V, respectively. Both ionization modes used a scan rate of 1.0 scan/s. For accurate mass correction, reference masses were continuously infused during all analyses: 121.0509 *m*/*z* (C_5_H_4_N_4_) and 922.0098 *m*/*z* (C_18_H_18_O_6_N_3_P_3_F_24_) in positive ionization mode, and 112.9856 *m*/*z* (C_2_O_2_F_3_(NH_4_)) and 1033.9881 *m*/*z* (C_18_H_18_O_6_N_3_P_3_F_24_) in negative ionization mode. Samples were analyzed in a single randomized run, with the instrument blank confirming no carryover. MS/MS analysis was performed using a collision energy of 20 eV. Data acquisition for MS and MS/MS was handled via the MassHunter Workstation software (version B.05.01, Agilent Technologies, Waldbronn, Germany).

### 3.4. Data Treatment

The raw data files were visually inspected using the MassHunter Qualitative Analysis Software (version B.07.00, Agilent Technologies) to detect background noises and unrelated ions. Background noise and unrelated ions were then removed with MassHunter Profinder (version B.10.00, Agilent Technologies) using the Batch Targeted Feature Extraction algorithm, reducing the data size and complexity. A custom manual database, based on the compound classification from the Phenol-Explorer database, was created and used as a reference during compound extraction. Key parameters included a peak threshold of >1000 counts, the grouping of isotopes and adducts into molecular features with a maximum charge of 2, and feature alignment within a 10 ppm ± 2 mDa mass and 0.15 min retention time window. All parameters were applied with a minimum frequency of 50% in at least one group. After integration, molecular characteristics were examined, and any inconsistencies were corrected manually. Following feature alignment, a logarithmic transformation of the data was performed prior to statistical analysis. Pre-processed data were then analyzed using pattern recognition methodologies, such as principal component analysis (PCA), a heatmap, pie charts, and volcano plots, to assess both expected and unexpected changes in features and identify comparable clusters.

### 3.5. Metabolite Identification

Compounds contributing significantly to class separation were identified through multiple methods, including tentative identification, LC-MS/MS fragmentation analysis, and, in some instances, a comparison with commercial standards such as gallic acid (Cat. No. G7384), quercetin-3-glucoside (Cat. No. Q4951), epicatechin (Cat. No. E1753), and p-coumaric acid (Cat. No. C9008), all from Sigma-Aldrich (St. Louis, MO, USA). Tentative identification was carried out by searching accurate mass data against public databases such as FooDB (https://foodb.ca/, accessed on 1 October 2024) and Phenol-Explorer (http://phenol-explorer.eu/, accessed on 1 October 2024), with a mass accuracy tolerance of 20 ppm for compound detection.

### 3.6. Determination of Antioxidant Capacity Using ABTS and FRAP

For the determination of the antioxidant activity, ABTS (2,2′-azinobis (3-ethylbenzothiazoline-6-sulfonic acid)) and FRAP (Ferric Reducing Antioxidant Power) methods were performed as referred [32]. A standard curve was prepared using Trolox dissolved in a MeO/H_2_O mixture acidified with 1% HCl at the following concentrations: 0, 0.125, 0.25, 0.5, 1, 2, and 4 µM. Assays were performed in three biological replicates and two technical for each sample. Absorbances were measured at 734 nm for ABTS and 595 nm for FRAP, using a microplate reader (PowerWave XS, Biotek, Saint-Jean-de-Védas, France). The results were expressed as μmol TE per g of fresh weight (FW), where TE refers to Trolox Equivalent.

### 3.7. Statistical Analyses

Statistical analysis was conducted using SPSS v.28.0 (IBM) to evaluate differences in antioxidant activity between samples. An independent samples *t*-test was performed, with a significance level of *p* < 0.05 considered indicative of statistically significant differences. Multivariate statistical analyses, including principal component analysis (PCA), were carried out using Unscrambler X 10.5. PCA was employed for unsupervised classification to examine the sample distribution within the study. A heatmap was constructed in R to visualize metabolite distribution patterns and the relative concentrations of flavonoids and non-flavonoids across different samples. To further characterize differential metabolite expression patterns, a volcano plot analysis was performed using MetaboAnalyst (v. 6.0). Additionally, a statistical test was conducted in R to compare the relative average composition of flavonoid and non-flavonoid classes from three biological replicates per group. A Student’s *t*-test was performed to assess significant differences between RiJ and RiS, with *p* < 0.05.

## 4. Conclusions

This study has provided an in-depth analysis of the phenolic composition, metabolic differences, and antioxidant capacity of “Regina” raspberries harvested in June (RiJ) and September (RiS). A targeted metabolomic analysis revealed significant differences between the two harvest periods, with RiS samples exhibiting higher concentrations of most flavonoid and non-flavonoid metabolites. Analytical methods, including PCA, heatmap, and volcano plot analyses, confirmed these metabolic differences, highlighting specific compounds such as cyanidin 3-sophoroside as the major anthocyanin present in this cultivar and hydroxybenzoic acid derivatives as compounds that vary significantly between RiJ and RiS. The antioxidant capacity of RiS raspberries was notably higher, as measured through ABTS and FRAP assays, correlating with their enriched phenolic profile. The predominance of bioactive compounds, such as anthocyanins, flavonoids, and phenolic acids, underscores the nutritional and commercial significance of RiS samples. These findings provide a solid foundation for developing postharvest treatments that extend the shelf life of late-harvest raspberries. They also offer a basis for further investigation into the functional roles of these metabolites in plant defense, stress resilience, and potential health benefits for humans.

## Figures and Tables

**Figure 1 plants-14-00888-f001:**
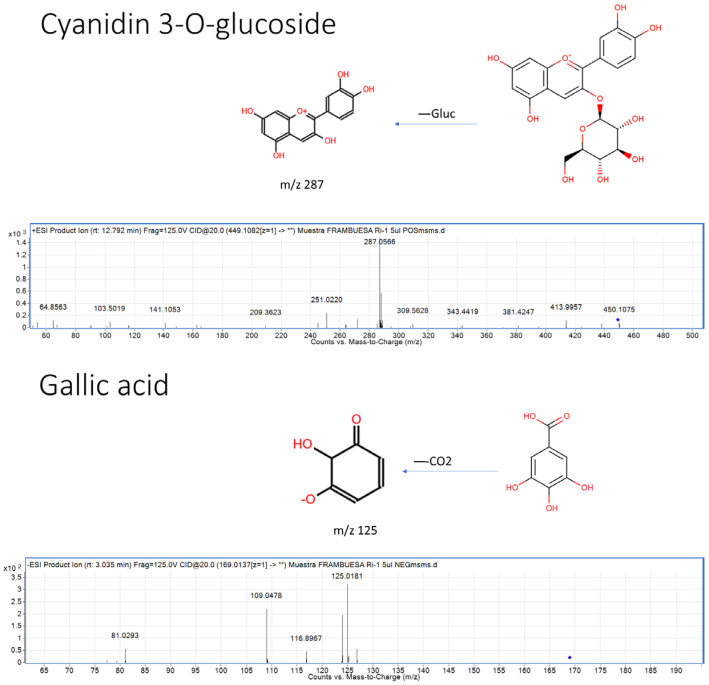
Examples of structures, fragmentation, and MS/MS spectra for the identification of two of the most representative polyphenols in raspberries according to the Phenol-Explorer database: an anthocyanin (cyanidin-3-glucoside) from the flavonoid group and a hydroxybenzoic acid (gallic acid) from the non-flavonoid group. The blue diamond (◆) indicates the precursor ion selected for fragmentation in the MS/MS analysis.

**Figure 2 plants-14-00888-f002:**
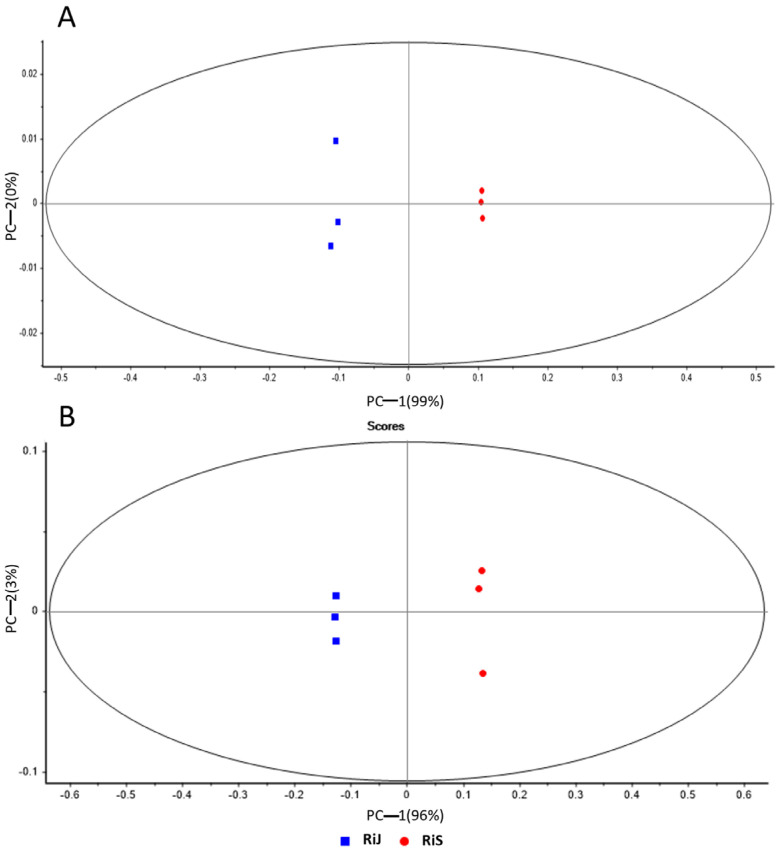
Principal component analysis of RiJ and RiS samples in (**A**) positive ionization mode and (**B**) negative ionization mode.

**Figure 3 plants-14-00888-f003:**
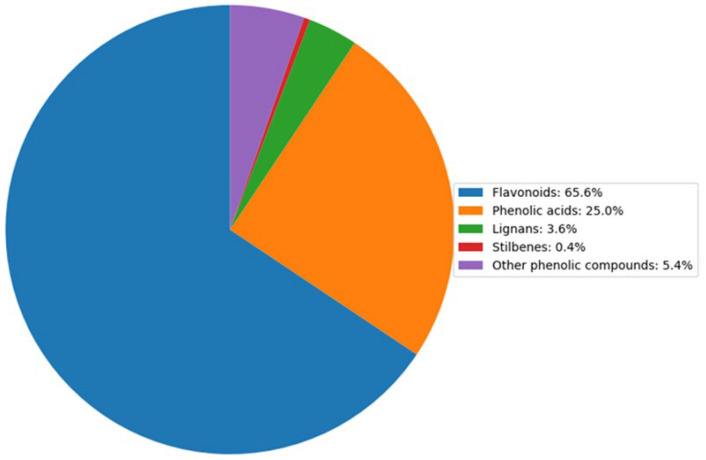
Distribution of polyphenol compounds in raspberry fruit (%). The pie chart illustrates the relative proportions of key polyphenol groups identified in raspberry samples. Data represent the average composition of three independent replicates analyzed using HPLC-MS, with values normalized to total polyphenol content.

**Figure 4 plants-14-00888-f004:**
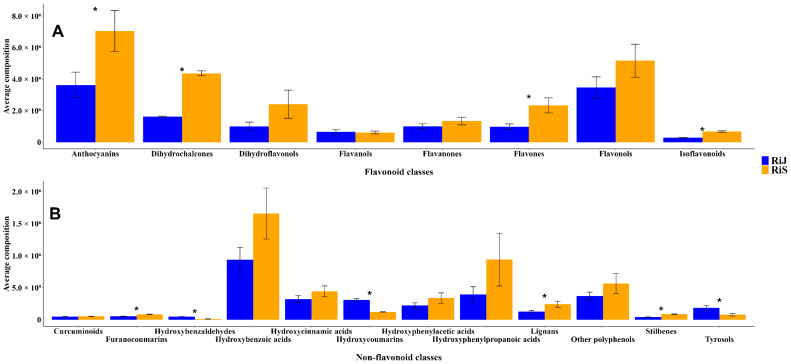
Average composition of (**A**) Flavonoids and (**B**) Non-flavonoids polyphenols present in RiJ and RiS samples. Values are expressed as means ± standard error (SEs), calculated from three biological replicates per group. Student’s *t*-test was performed to assess significant differences between RiJ and RiS for each compound class. Asterisks (*) indicate statistically significant differences with *p* < 0.05.

**Figure 5 plants-14-00888-f005:**
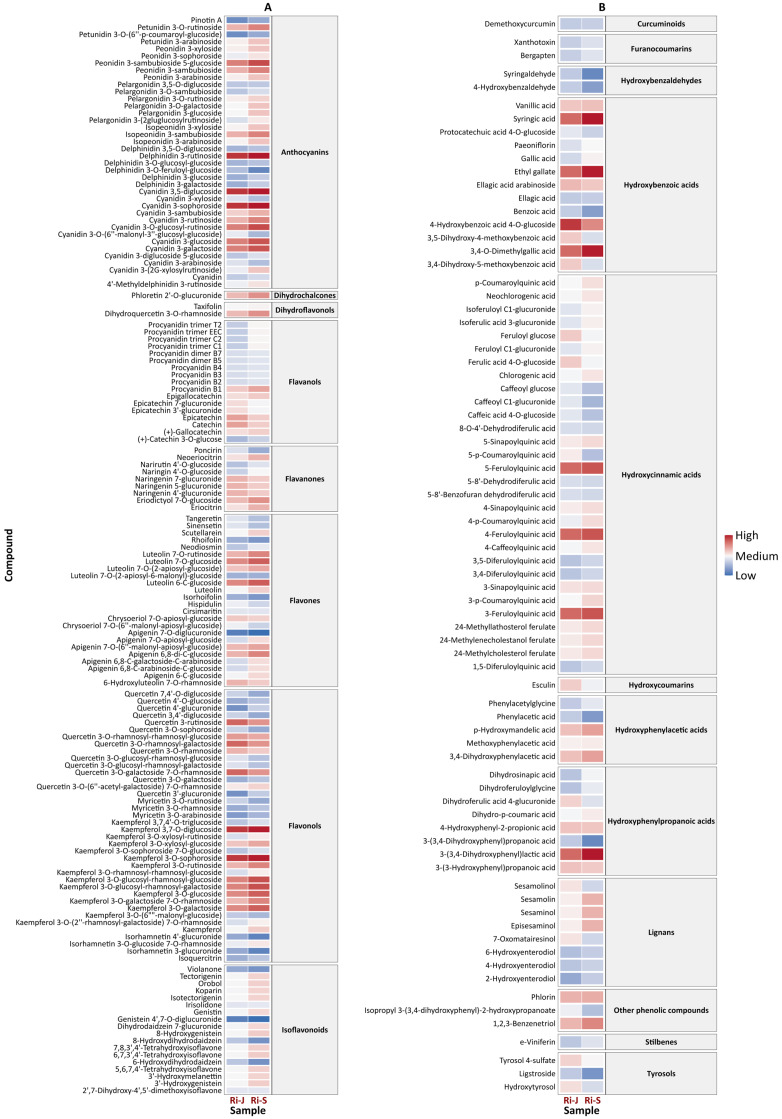
Heatmap of (**A**) flavonoids and (**B**) non-flavonoids in RiJ and RiS samples. The heatmap displays the relative abundance of key flavonoids and polyphenols in raspberries harvested at two different periods (RiJ and RiS). The rows represent individual polyphenols, while the columns correspond to RiJ or RiS. The color gradient indicates relative concentrations, with red denoting higher levels and blue indicating lower levels.

**Figure 6 plants-14-00888-f006:**
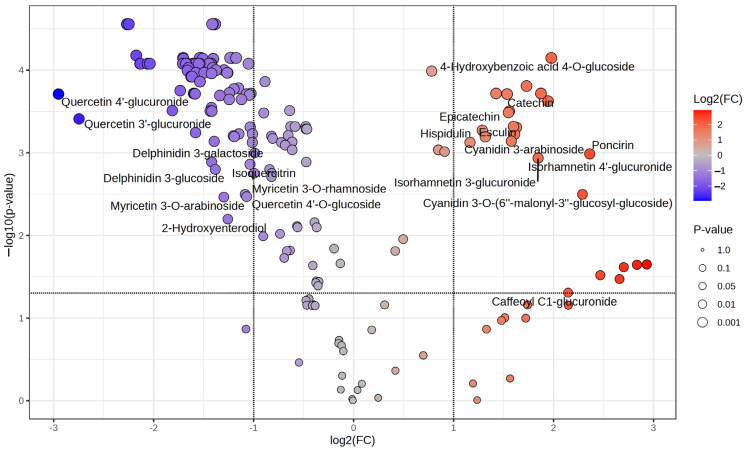
Volcano plot showing the differential abundance of metabolites in raspberry samples. Each point represents a metabolite, plotted based on its log_2_ fold change (*x*-axis) and −log_10_(*p*-value) (*y*-axis). The further its position is away from the (0,0), the more significant the feature is. The red circles represent features above the threshold, indicating significantly increasing metabolites, while the blue circles represent significantly decreasing metabolites. The gray points denote metabolites with non-significant changes. Key polyphenols are labeled for reference.

**Figure 7 plants-14-00888-f007:**
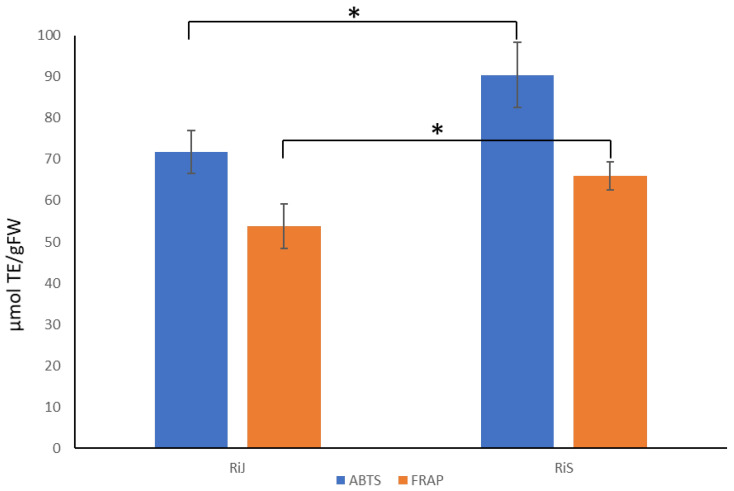
Antioxidant capacity determined via ABTS and FRAP in RiJ and RiS samples. Values are means ± SDs; n = 3. The asterisk indicates that samples are statistically different according to Student’s *t*-test (*p* < 0.05).

## Data Availability

Data are contained within the article or Appendix A.

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
