# Peer review of "Metabolic and Antioxidant Variations in “Regina” Raspberries: A Comparative Analysis of Early and Late Harvests"

_plants, 2025, doi:10.3390/plants14060888_

Round 1
Reviewer 1 Report
Comments and Suggestions for Authors
Line 31 Not all raspberry varieties are European (R. idaeus var idaeus). American raspberries such as Heritage and Summit have glandular hairs and are designated as R. idaeus var. strigosus
Line 61 anthocyanins are not higher in unripe raspberries.
Fig 4b y axis names are not aligned with bars.
Figure 5. Cannot read the vertical axis. When magnified, text becomes pixelated.
Supplementary table is helpful but a major pigment, cyanidin-3-sophoroside, associated with raspberry is not in this table. Why is it not in ‘Regina’? I see cyanidin-3-sophorose (sophoroside?) mentioned in the conclusion but not seeing it elsewhere.
Several other anthocyanins are identified, such as delphinidin-3-glucoside and delphinidin-3-galactoside, which are not identified in other raspberry pigment studies. What are the relevant contents? Is it possible that these pigments could contribute to the purpling that often occurs in some varieties?
There are a number of new compounds identified in this paper. It would be helpful if some discussion was provided about these compounds, and especially the relative content of these compared to the more commonly identified compounds.
This citation provides some insight and also some early data on seasonal differences in raspberry
Kassim, A., Poette, J., Paterson, A., Zait, D., McCallum, S., Woodhead, M., Smith, K., Hackett, C. and Graham, J., 2009. Environmental and seasonal influences on red raspberry anthocyanin antioxidant contents and identification of quantitative traits loci (QTL). Molecular nutrition & food research, 53(5), pp.625-634.
Author Response
Dear Reviewers,
First and foremost, we would like to express our sincere gratitude for the thoughtful, detailed comments and constructive criticism, which have significantly improved the manuscript. Below, we provide our responses to the comments.
(Note: Reviewer comments are in italics; authors' responses are in regular text.)
The manuscript has been updated with all the corrections, followed by the accepted changes to facilitate readability.
Reviewer 1
Line 31 Not all raspberry varieties are European (R. idaeus var idaeus). American raspberries such as Heritage and Summit have glandular hairs and are designated as R. idaeus var. Strigosus
-Thank you for the comment. In the text, Rubus idaeus was mentioned in a general sense, encompassing both European and American varieties. However, we agree that it is important to clarify the specific type, as mentioning only the European varieties could lead to confusion. Therefore, the paragraph has been revised for clarification.
Line 61 anthocyanins are not higher in unripe raspberries.
-Thank you for the comment. We agree with the reviewer that anthocyanins are not higher in unripe raspberries. As pointed out, this could lead to misunderstandings. The paragraph has been revised in the manuscript, and an additional citation has been included to clarify this point. Consequently, the reference numbers have been updated throughout the manuscript.
Fig 4b y axis names are not aligned with bars.
-Thank you for the comment. We have improved the Figure 4b in order to align the names and the bars. We have included also the error bars and significance.
Figure 5. Cannot read the vertical axis. When magnified, text becomes pixelated.
-Thank you for the comment. We have improved the Figure 5 to make it more readable.
Supplementary table is helpful but a major pigment, cyanidin-3-sophoroside, associated with raspberry is not in this table. Why is it not in ‘Regina’? I see cyanidin-3-sophorose (sophoroside?) mentioned in the conclusion but not seeing it elsewhere.
-Thank you for the comment. First of all, the term cyanidin-3-sophorose has been changed throughout the manuscript to cyanidin-3-sophoroside as it was a mistake. Regarding the question of why this metabolite is not included in the table, it is because Supplementary Table S1 lists all the significant metabolites between the two conditions, RiJ and RiS samples, derived from the volcano plot analysis.
Several other anthocyanins are identified, such as delphinidin-3-glucoside and delphinidin-3-galactoside, which are not identified in other raspberry pigment studies. What are the relevant contents? Is it possible that these pigments could contribute to the purpling that often occurs in some varieties?
-Thank you for your comments. In our study, the primary goal was to investigate differences in metabolic compounds and antioxidant capacity between the harvest periods, rather than to quantify the phenolic compounds within each harvest. This is why we did not provide specific measurements of each metabolite and why some compounds, which are commonly abundant in both groups, are not presented in the figures.
Regarding delphinidin-3-galactoside, we acknowledge that this was mistakenly mentioned in the paper. This error arose from the mixing of two parallel datasets—one involving raspberries and the other involving blueberries. We apologize for this oversight. We have checked the fragmentation profile in both matrices and confirmed that it was not part of the raspberry profile. We have made the necessary corrections in the manuscript.
Delphinidin-3-O-glucoside is one of the 11 common anthocyanins present in berries (cranberry, elderberry, and mulberry) and grains (black rice and black soybean) studied by Xie et al., 2024. Additionally, a study by Ponder et al., (2021) on raspberries, blackberries, red currants, blackcurrants, and highbush blueberries also reports the presence of Del-3-Glu in raspberries and Del-3-Gal in blueberries. Delphinidin derivatives, including delphinidin-3-glucoside and delphinidin-3-galactoside, are known for producing purple to blue hues. While raspberry varieties generally have more abundant anthocyanins like cyanidin-based compounds (which contribute to red colors), the presence of delphinidin-related anthocyanins, even in small amounts, can contribute to a purplish or bluish tint, especially under certain environmental conditions or during specific stages of ripening.
Xie B, Wang M, Yang D. Identification of anthocyanins in deep colored berries and grains in China. Food Chem X. 2024 Jun 27;23:101602. doi: 10.1016/j.fochx.2024.101602
Ponder, et al. (2021). Genetic Differentiation in Anthocyanin Content among Berry Fruits. Curr. Issues Mol. Biol. 2021, 43, 36–51. https://doi.org/10.3390/cimb43010004
There are a number of new compounds identified in this paper. It would be helpful if some discussion was provided about these compounds, and especially the relative content of these compared to the more commonly identified compounds.
Thank you for your valuable comment. Our analysis is based on relative abundance rather than absolute quantification, as no calibration curves were established for the novel compounds. Therefore, while we can compare the relative intensities of these newly identified compounds with those of more commonly known flavonoids and non-flavonoids, these values should be interpreted as relative estimates rather than precise concentrations.
Regarding a more detailed discussion of their biological roles and functional implications, given that these compounds are novel in raspberry, we consider their study to be an interesting avenue for future research, and we will take your suggestion into account.
This citation provides some insight and also some early data on seasonal differences in raspberry
Kassim, A., Poette, J., Paterson, A., Zait, D., McCallum, S., Woodhead, M., Smith, K., Hackett, C. and Graham, J., 2009. Environmental and seasonal influences on red raspberry anthocyanin antioxidant contents and identification of quantitative traits loci (QTL). Molecular nutrition & food research, 53(5), pp.625-634.
-Thank you for this Reference. It has been included in the Introduction section to support the variations observed in the quantification of phenolic compounds which can be influenced by factors such as cultivation conditions, processing methods, and seasonal variations, among others.

Reviewer 2 Report
Comments and Suggestions for Authors
This paper investigates the differences in metabolic and antioxidant properties of ‘Regina’ red raspberries at two different harvesting periods (June and September), and the effect of harvesting time on the polyphenol content and antioxidant capacity was revealed by comparative analyses. But in my opinion, the overall research content of the article is a bit simple, including 1) the number of samples is small, containing only fruit materials of a single variety and a single harvesting time point; 2) the analysis method is simple, using only metabolome for compound identification and quantification; 3) the analysis content is a bit thin, and only the differential metabolites in the two groups of samples were analysed and identified, lacking the exploration of deeper differential mechanisms. For example, which synthetic key genes in the metabolic pathway with high expression or low expression of degradation-related genes led to high accumulation of phenolic compounds in RiS samples? Or which environmental factors in the two growing seasons led to higher phenolic accumulation in RiS fruits, etc. Therefore, the article is slightly less innovative. Many previous articles have identified polyphenolic compounds in raspberries and analysed the main antioxidant active substances,so it is suggested to add some investigations on the mechanism or the study of post-harvest storage process in order to enrich the research content and novelty of the article.
Major issues:
- Figure 1 on the analysis of cyanidin-3-glucoside and gallic acid only shows the results of the identification of the two substances and does not analyse the difference in the content of the two substances in the two sets of samples. It is suggested to add.
- 2. What is the meaning of the ‘most representative polyphenol’in Line 93? Highest abundance? The characteristic MS fragments of the compounds identified using MS/MS are suggested to be listed, such as cyanidin-3-glucoside, gallic acid, etc. In addition, no quantitative analysis of compounds was performed in the article, so the calibration quantification or internal standard quantification of important differential metabolites is suggested.
- The discussion about the results is not sufficient. For example, Line108-110 mentioned that the effects of different harvesting seasons on the compound fractions of 10 raspberries were analysed in a previous study (Mazuret al., 2014), and it is suggested to briefly describe the main results, especially the differences of polyphenolic compounds in different post-harvest seasons, and compare them with the results of this study.
- It is not clear whether the proportions shown in Figure 3 are the percentage of the number of compounds or the percentage of the content, which is not clearly expressed. s the proportion of 224 metabolites the result of the combination of the two groups of samples? It is suggested to show the differences in the types of compounds identified in the two groups separately.
- The concentrations of each class of compounds presented in Figure 4 need to include standard errors of 3 biological replicates and be analyzed for statistical significance of difference.
- Figure 5 illustrates a heatmap of the content of each class of compounds, but does not cluster the compounds and does not conform to the hierarchical cluster analysis of the title. It is suggested here that the volcano diagram analysis of Figure 6 should be carried out first to screen out the differential metabolites, and then the differential metabolites should be subjected to clustering analysis, so that it can be analyzed whether the same class of metabolites from the same branch metabolic pathway have the same trend of change, so as to explore the molecular mechanism in more depth.
- The presentation of Figure 7 is not more reasonable, and it is suggested to set the horizontal coordinates to two antioxidant activities instead of two groups of samples. In addition why n=6 in Figure 7, if test replicates are included, the mean value of test replicates should be used as the result of biological replicates.
Minor issues:
- In the abstract, it is mentioned that ‘377 unique metabolites were identified and categorised’, which is inconsistent with ‘After eliminating duplicated compounds from both ionisation modes, 224 unique metabolites were identified’in Line 116 in the results section.
- The plant material section should explain how the biological replicates were set up and how many fruits were included in each biological replicate.
- In Line 351-353, ‘Compounds contributing significantly to class separation were identified through multiplemethods, including tentative identification, LC-MS/MS fragmentation analysis, and, in some instances, by comparison with commercial standards.’ The characteristic MS fragments of these substances should be given in the supplemental table, and which substances were identified by the commercial standards comparison should be listed.
- The public database for compound identification suggests attached link, such as FooDB and Phenol Explorer. In addition, in Line 355, why the mass accuracy difference choose 20ppm, ppm was the concentration unit, I do not understand.
- What is the wavelength of absorbance in Line 362.
- In line129-130, two sentences ‘flavanol concentrations were comparable between RiJ and RiS.’are repeated.
Comments on the Quality of English Language
The English could be improved to more clearly express the research.
Author Response
Dear Reviewers,
First and foremost, we would like to express our sincere gratitude for the thoughtful, detailed comments and constructive criticism, which have significantly improved the manuscript. Below, we provide our responses to the comments.
(Note: Reviewer comments are in italics; authors' responses are in regular text.)
The manuscript has been updated with all the corrections, followed by the accepted changes to enhance readability.
Reviewer 2
This paper investigates the differences in metabolic and antioxidant properties of ‘Regina’ red raspberries at two different harvesting periods (June and September), and the effect of harvesting time on the polyphenol content and antioxidant capacity was revealed by comparative analyses. But in my opinion, the overall research content of the article is a bit simple, including 1) the number of samples is small, containing only fruit materials of a single variety and a single harvesting time point; 2) the analysis method is simple, using only metabolome for compound identification and quantification; 3) the analysis content is a bit thin, and only the differential metabolites in the two groups of samples were analysed and identified, lacking the exploration of deeper differential mechanisms. For example, which synthetic key genes in the metabolic pathway with high expression or low expression of degradation-related genes led to high accumulation of phenolic compounds in RiS samples? Or which environmental factors in the two growing seasons led to higher phenolic accumulation in RiS fruits, etc. Therefore, the article is slightly less innovative. Many previous articles have identified polyphenolic compounds in raspberries and analysed the main antioxidant active substances,so it is suggested to add some investigations on the mechanism or the study of post-harvest storage process in order to enrich the research content and novelty of the article.
Major issues:
Figure 1 on the analysis of cyanidin-3-glucoside and gallic acid only shows the results of the identification of the two substances and does not analyse the difference in the content of the two substances in the two sets of samples. It is suggested to add.
-Thank you for the comment. In the Figure 1, the aim was to show how the identification of metabolites was performed rather to quantify them. That is the reason why one example from the flavonoid and one example from the non-flavonoid group was selected. Furthermore, along the manuscript, relative contents are shown for all the metabolites, and no data of quantification are given.
- 2. What is the meaning of the ‘most representative polyphenol’in Line 93? Highest abundance? The characteristic MS fragments of the compounds identified using MS/MS are suggested to be listed, such as cyanidin-3-glucoside, gallic acid, etc. In addition, no quantitative analysis of compounds was performed in the article, so the calibration quantification or internal standard quantification of important differential metabolites is suggested.
We appreciate your insightful feedback and the opportunity to improve the manuscript. When referring to the "most representative polyphenol," we mean compounds that have been previously reported in the literature and databases such as Phenol-Explorer as major constituents in raspberry, rather than those with the highest relative abundance in our dataset. In response to your suggestion, we have included a small table listing some of the most commonly reported polyphenols (Supplementary Table S1), along with their characteristic MS/MS fragments obtained from our experimental samples for greater clarity.
The discussion about the results is not sufficient. For example, Line108-110 mentioned that the effects of different harvesting seasons on the compound fractions of 10 raspberries were analysed in a previous study (Mazuret al., 2014), and it is suggested to briefly describe the main results, especially the differences of polyphenolic compounds in different post-harvest seasons, and compare them with the results of this study.
-Thank you for your suggestion. The revised manuscript has been improved by adding some discussion related to this issue, as the reviewer proposed. However, since in this part of the article the aim of the PCA is to highlight the huge separation between samples in terms of metabolite composition due to harvesting periods and, at this point of the article, no metabolite is yet mentioned, we do not find necessary to discuss the metabolites quantified in the article with our results.
It is not clear whether the proportions shown in Figure 3 are the percentage of the number of compounds or the percentage of the content, which is not clearly expressed. Is the proportion of 224 metabolites the result of the combination of the two groups of samples? It is suggested to show the differences in the types of compounds identified in the two groups separately.
-Thank you for your comment. In Figure 3, the proportions represent the percentage of identified compounds within each phenolic class, not their concentrations. The pie chart was constructed by classifying all identified phenolic compounds from both flowering periods (June and September) and calculating the proportion that each class represents relative to the total number of identified compounds (224 metabolites). To clarify, this representation does not reflect the quantitative content of each class, but rather their relative diversity within the sample. This approach was chosen to provide an overview of the dominant phenolic classes in raspberries, regardless of their absolute concentrations.
The concentrations of each class of compounds presented in Figure 4 need to include standard errors of 3 biological replicates and be analyzed for statistical significance of difference.
-Thank you for your suggestion. We have revised Figure 4 to include error bars and statistical significance. The results and discussion related to this Figure have been improved. Furthermore, data related to the statistical analysis have been included in the M&M section.
Figure 5 illustrates a heatmap of the content of each class of compounds, but does not cluster the compounds and does not conform to the hierarchical cluster analysis of the title. It is suggested here that the volcano diagram analysis of Figure 6 should be carried out first to screen out the differential metabolites, and then the differential metabolites should be subjected to clustering analysis, so that it can be analyzed whether the same class of metabolites from the same branch metabolic pathway have the same trend of change, so as to explore the molecular mechanism in more depth.
Thank you for your insightful comment. You are absolutely right that Figure 5 does not present a hierarchical clustering analysis, and we have corrected the description to accurately reflect that it is a heatmap visualization rather than a clustering analysis.
The primary objective of this figure is to provide a clear representation of the relative concentrations of flavonoids across different samples, categorized by metabolite class, without applying hierarchical clustering algorithms. The compounds are displayed according to their predefined classification rather than being reorganized based on similarity metrics.
Additionally, we made this decision to avoid redundancy in visual information, considering that Figure 6 already presents a volcano plot, which effectively highlights the most significant differential compounds. Given that the volcano plot allows for a more focused identification of key metabolites, adding hierarchical clustering in Figure 5 could have resulted in an overlap of information rather than providing distinct insights.
The presentation of Figure 7 is not more reasonable, and it is suggested to set the horizontal coordinates to two antioxidant activities instead of two groups of samples. In addition, why n=6 in Figure 7, if test replicates are included, the mean value of test replicates should be used as the result of biological replicates.
-Thank you for the comment. Certainly, we were considering how to present the results, either as they are or as suggested by the reviewer. However, since the goal is to emphasize the difference in antioxidant capacity between the two groups based on harvest time, we found it more logical to keep it as it is, as it better supports the interpretation of the results. If we place the two antioxidant capacity detection methods on the x-axis, more emphasis would be placed on the method used rather than the differences between the groups. Therefore, we believe it is more appropriate to retain the original format in the manuscript. Additionally, as suggested by the reviewer, there was an error regarding the number of replicates. The correct value is n=3, as three biological replicates were used for the determination, with each biological sample consisting of a mixture of pulverized fruits frozen with liquid nitrogen. We have corrected this mistake in the revised manuscript.
Minor issues:
In the abstract, it is mentioned that ‘377 unique metabolites were identified and categorised’, which is inconsistent with ‘After eliminating duplicated compounds from both ionisation modes, 224 unique metabolites were identified’ in Line 116 in the results section.
-Thank you for your comment. The word “unique” has been removed from the Abstract section, as the 377 metabolites included duplicates and were not unique, unlike the 224 metabolites mentioned later.
The plant material section should explain how the biological replicates were set up and how many fruits were included in each biological replicate.
-Thank you for the comment. As indicated in the Plant Material section, “Some fruits were analysed for quality assessments, while the rest were frozen in liquid nitrogen, ground into a fine powder, and stored at -80ºC until analyses were performed”. For the HPLC analyses and for the antioxidant determinations, three biological replicates were used. Each biological replicate consisted of 0.2 g of a homogeneous mixture of frozen raspberries in fine powder form. This is also indicated in the Extraction Procedure section.
In Line 351-353, ‘Compounds contributing significantly to class separation were identified through multiple methods, including tentative identification, LC-MS/MS fragmentation analysis, and, in some instances, by comparison with commercial standards.’ The characteristic MS fragments of these substances should be given in the supplemental table, and which substances were identified by the commercial standards comparison should be listed.
-Thank you for your comment. In response to your suggestion, we have included a small table listing some of the most commonly reported polyphenols, along with their characteristic MS/MS fragments obtained from our experimental samples for greater clarity. In M&M it has been included which one correspond to the standard commercials.
The public database for compound identification suggests attached link, such as FooDB and Phenol Explorer. In addition, in Line 355, why the mass accuracy difference choose 20ppm, ppm was the concentration unit, I do not understand.
Thank you for your comment. In this context, ppm refers to mass accuracy tolerance, not concentration units. It defines the allowable deviation between the measured and theoretical mass of a compound in mass spectrometry.
The Agilent 6530 Q-TOF used in this study has a mass resolution of 10,000–20,000 FWHM at m/z 200, which is considered moderate compared to higher-end Q-TOF instruments. Based on this resolution, a tolerance of 20 ppm was chosen as a balance between specificity and sensitivity in metabolite identification. A stricter tolerance (e.g., 5–10 ppm) might reduce false positives but could also exclude valid metabolite matches due to small instrumental mass deviations. In contrast a wider tolerance could lead to incorrect assignments by allowing too many potential matches.
Thus, 20 ppm was selected as an optimal compromise, ensuring reliable metabolite identification while accounting for minor mass variations inherent to the instrument. This approach aligns with standard practices in metabolomics using mid-range Q-TOF systems.
By other side, the links to the databases used are now included in the M&M section.
What is the wavelength of absorbance in Line 362.
-Thank you for the comment. The information regarding the wavelengths (734 nm for ABTS and 595 nm for FRAP) has been added to the M&M section.
In line129-130, two sentences ‘flavanol concentrations were comparable between RiJ and RiS.’are repeated.
-Thank you for the note, this has been corrected in the revised manuscript.

Reviewer 3 Report
Comments and Suggestions for Authors
The article ‘Metabolic and Antioxidant Variations in 'Regina' Raspberries: A Comparative Analysis of Early and Late Harvests ‘ presented for review is interesting. However, some issues need to be clarified or supplemented. The comments are included below:
Abstract
- Line 12-15: It is common practice to standardize research procedures in order to compare results. Writing about the influence of extraction on the content of compounds is partly correct but also highly controversial. Please rewrite this fragment.
- Line 21-24: The statement that post-harvest treatments are necessary to maintain the quality of raspberries is not very clear and requires further clarification.
Introduction
- Line 35-36: Which country is the fifth leading producer of raspberries?
- Line 61-62: Please cite other literature that would support the thesis that anthocyanins are present in much higher concentrations in unripe. In my opinion, this statement is highly controversial and requires deeper justification. One cannot rely on one article in this regard.
- At the end of the introduction the purpose of the work should be clearly defined.
Results and Discussion
- Figure 5: Poor legibility of the drawing. The descriptions in Figure 5 are difficult to read. Please correct them.
Material and Methods
3.2. Extraction Procedure
- Line 299-301: Please provide the methodology according to which the extraction was performed.
- Please also provide the literature in which such low concentrations of methanol are used for extraction.
- Please also provide the methodology in which such a short extraction is used instead of multiple extraction to decolorize the precipitate.
Author Response
Dear Reviewers,
First and foremost, we would like to express our sincere gratitude for the thoughtful, detailed comments and constructive criticism, which have significantly improved the manuscript. Below, we provide our responses to the comments.
(Note: Reviewer comments are in italics; authors' responses are in regular text.)
The manuscript has been updated with all the corrections, followed by the accepted changes to enhance readability.
Reviewer 3
The article ‘Metabolic and Antioxidant Variations in 'Regina' Raspberries: A Comparative Analysis of Early and Late Harvests` presented for review is interesting. However, some issues need to be clarified or supplemented. The comments are included below:
Abstract
- Line 12-15: It is common practice to standardize research procedures in order to compare results. Writing about the influence of extraction on the content of compounds is partly correct but also highly controversial. Please rewrite this fragment.
-Thank you for the comment. The fragment has been rewritten to avoid misunderstandings: “However, their extraction and quantification are influenced by factors such as genetic variability, environmental conditions, fruit maturity, storage, and processing techniques”
- Line 21-24: The statement that post-harvest treatments are necessary to maintain the quality of raspberries is not very clear and requires further clarification.
-Thank you for the comment. It is well known that raspberries are highly perishable fruits. To extend their postharvest life, they need to be refrigerated and/or subjected to various postharvest treatments, such as low temperatures, CO2 gas treatments, packaging under modified atmospheres, etc. Understanding the differences between harvest periods of a cultivar helps modulate the levels of certain compounds. For example, low temperatures are known to increase anthocyanin content. Our research group specializes in applying short CO2 treatments to fruits to maintain quality and extend their storage life. Therefore, additional manuscripts examining the effect of low temperature and the application of high levels of CO2 for short periods on modulating some of these metabolites are currently in progress. Since this is part of the Abstract section, and to avoid misunderstandings, we have removed the sentence “necessary to maintain the quality of raspberries” and improved the last paragraph of the Introduction section.
Introduction
- Line 35-36: Which country is the fifth leading producer of raspberries?
- Thank you for the comment. According to 2022 data, Mexico is the fifth leading producer of raspberries, and this information has been included in the Introduction section. These five countries have been the top producers since at least 2015.
- Line 61-62: Please cite other literature that would support the thesis that anthocyanins are present in much higher concentrations in unripe. In my opinion, this statement is highly controversial and requires deeper justification. One cannot rely on one article in this regard.
- Thank you for the comment. We agree with the reviewer that this statement could lead to misunderstandings. We have clarified that the mentioned increase in anthocyanin levels in unripe fruit was an exception for a specific cultivar. The paragraph has been revised in the manuscript, and an additional reference has been added. As a result, the reference numbers have been updated throughout the manuscript.
- At the end of the introduction the purpose of the work should be clearly defined.
-Thank you for your comment. The revised manuscript has been improved by adding more information about the purpose of the work at the end of the Introduction section.
Results and Discussion
- Figure 5: Poor legibility of the drawing. The descriptions in Figure 5 are difficult to read. Please correct them.
-Thank you for the comment. We have improved the quality of Figure 5 to ensure that the descriptions are readable.
Material and Methods
3.2. Extraction Procedure
- Line 299-301: Please provide the methodology according to which the extraction was performed.
- Please also provide the literature in which such low concentrations of methanol are used for extraction.
- Please also provide the methodology in which such a short extraction is used instead of multiple extraction to decolorize the precipitate.
-Thank you for the comments. We have included additional details in the extraction procedure in the revised manuscript. Regarding your suggestions, the extraction method described in M&M section (3.2) has been previously used to analyze total phenolic and total anthocyanin content, as well as to determine antioxidant capacity in other soft fruits such as blueberries, yielding reliable results (Foods 2023, 12, 2621. https://doi.org/10.3390/foods12132621).
We believe that the 50:50 ratio is effective for extracting polar compounds, which are abundant in these samples. Additionally, our extraction is performed in two repetitions until 2 mL of supernatant is obtanied, with a 2-hour incubation. Similarly, Saafi et al. (2009) used 50% methanol as a solvent to extract phenolic compounds from plants at room temperature. Their extraction process was assisted by an orbital shaker at a frequency of 200 rpm for 2hours, followed by centrifugation at 1000g for 15min. As a result, the samples they tested were found to be rich in total phenolics, ranging from 209.42 to 447.73 mg equivalent gallic acid⁄100 g (GAE/100 g) fresh weight.
It is well known that the extraction efficiency of conventional methods depends on the extraction duration within a certain time range. A longer extraction duration increases efficiency; however, once solute equilibrium is reached between the solid and liquid phases, extending the time will not longer improve extraction To achieve a higher yield, a greater solvent-to-solid ratio is required, but when the ratio becomes too high, excessive solvent is used, and extraction time increases. As a result, conventional extraction methods, which typically use organic solvent maceration, require large volumes of solvent and long extraction times, making them stable but not always efficient enough (Shi et al., 2022)
We agree with the reviewer, that increasing the extraction time could enhance the extraction of phenolic compounds. Considering additional extractions such as conducting a second extraction using 70% acetone to extract flavanols, could improve extration efficiency and better recover phenolic compounds. However, we believe that our extraction procedure is sufficient to demonstrate the differences in phenolic compunds between the two harvest periods, as intended in this manuscript.
Sanchez-Ballesta et al., 2023. Are the Blueberries We Buy Good Quality? Comparative Study of Berries Purchased from Different Outlets. Foods 2023, 12, 2621. Doi:10.3390/foods12132621
Saafi et al., 2009. Phenolic content and antioxidant activity of four date palm (Phoenix dactylifera L.) fruit varieties grown in Tunisia. Int J Food Sci Technol. 44(11):2314–2319. Doi: 10.1111/j.1365-2621.2009.02075.x
Shi et al., 2022. Extraction and characterization of phenolic compounds and their potential antioxidant activities. Environ Sci Pollut Res Int. 2022, 29(54):81112–81129. Doi: 10.1007/s11356-022-23337-6

Reviewer 4 Report
Comments and Suggestions for Authors
The manuscript ,,Metabolic and Antioxidant Variations in 'Regina' Raspberries: A Comparative Analysis of Early and Late Harvests" (plants-3470564) is interesting and within the scope of the journal.
The introduction provides sufficient background and references.
The word terpenes must be deleted from line 53 because it appears in line 52.
The research design is in line with the objectives of the study and the methods are described in an adequate way.
The study provides an in-depth analysis of the phenolic composition, metabolic differences, and antioxidant capacity of 'Regina' primocane raspberries harvested in June (RiJ) and September (RiS). Targeted metabolomic analysis of 'Regina' primocane raspberries revealed significant differences between the two harvest periods, with RiS samples showing higher concentrations of most flavonoid and non-flavonoid metabolites. These metabolic differences, which highlight specific compounds from 'Regina' raspberries, have been confirmed by Principal Component Analysis (PCA), Hierarchical Cluster Analysis (HCA) and Volcano Plot analyses.
The conclusions are supported by the results, which are presented in an organised manner with figures and detailed descriptions.
Author Response
Dear Reviewers,
First and foremost, we would like to express our sincere gratitude for the thoughtful, detailed comments and constructive criticism, which have significantly improved the manuscript. Below, we provide our responses to the comments.
(Note: Reviewer comments are in italics; authors' responses are in regular text.)
The manuscript has been updated with all the corrections, followed by the accepted changes to enhance readability.
Reviewer 4
The manuscript “Metabolic and Antioxidant Variations in 'Regina' Raspberries: A Comparative Analysis of Early and Late Harvests" (plants-3470564) is interesting and within the scope of the journal.
The introduction provides sufficient background and references.
The word terpenes must be deleted from line 53 because it appears in line 52.
The research design is in line with the objectives of the study and the methods are described in an adequate way.
The study provides an in-depth analysis of the phenolic composition, metabolic differences, and antioxidant capacity of 'Regina' primocane raspberries harvested in June (RiJ) and September (RiS). Targeted metabolomic analysis of 'Regina' primocane raspberries revealed significant differences between the two harvest periods, with RiS samples showing higher concentrations of most flavonoid and non-flavonoid metabolites. These metabolic differences, which highlight specific compounds from 'Regina' raspberries, have been confirmed by Principal Component Analysis (PCA), Hierarchical Cluster Analysis (HCA) and Volcano Plot analyses.
The conclusions are supported by the results, which are presented in an organised manner with figures and detailed descriptions.
-Thank you for your valuable comments and feedback on our manuscript, In response to your suggestion regarding the term "terpenes" on line 53, we have carefully reviewed the text and deleted the word as it appears in line 52, as you recommended.

Round 2
Reviewer 1 Report
Comments and Suggestions for Authors
Authors have made numerous changes and improvements to the ms. There are some additional changes that need to be made.
Line 63, 2012 is not exactly recent. There do not seem to be clear statistics on the percent of the raspberry industry used for fresh markets, although it appears that the 15% global raspberry production from Mexico is 90% fresh market and most of US production is also fresh (see https://www.usitc.gov/publications/332/pub5194.pdf). Not sure if the no. 2 reference gave a guesstimate of fresh market raspberry; I would say 20% of global production is fresh market but most likely centered in the Americas.
Line 65, go back to Tan and refer to the FAO reference they used. Currently the sentence used here is close to word to word from Tan 2022. Line 66, Mexico seems to be second to Russia at the moment, looking at other sources. It might be better to state the top 5 world leaders in red raspberry production are Russa, Serbia, Mexico, United States, Poland.
Line 71, replace ‘The varieties’ with ‘genotypes’.
Line 55 and line 80 are almost the same. Suggest changing line 80.
Line 84 research does not identify; those doing the research do this.
Lines 93-98. R. chingii is different from red raspberry, apparently because carotenoids increase with ripening and provide most of the red color. This addition confuses the point of the present experiment. Anthocyanins in most berries increase, but not all flavonoids increase with ripening. Here, the point is to utilize newer technologies to follow seasonal effects on flavonoid profiles of a newer primocane variety.
Line 120, it’s a pretty big stretch to move to metabolites-postharvest, for a couple of reasons. One is that different temperatures during production and harvest (field heat) can mitigate specific metabolic profiles. This sentence (L 472-473) in the conclusion needs to be removed.
Methods: were internal standards used?
Line 180 implies several studies showed higher flavonoids in later harvests-references? Line 194, why would tyrosols and hydroxycoumarins be higher in the RiJ samples?
Figure 4-need to add in A, B and increase font on y axis, also enlarge legend.
Fig 5, still can’t make out the y axis names. Seems to be partly a font and partly an uneven black color.
Lines 224-243 need to be better singulated between plant pathogens and human bioactives.
Line 292 should start a new paragraph. Lines 298-304 are also hard to follow-a lot of ‘this’ which makes it difficult to follow what is alluded to, and wandering sentences. Different plateaus-what are these?
Line 337-maturity index is not helpful here. Raspberries are 8-12% ssc and 0.8-1.2% titratable acidity. These ratios of 4 and 3 indicate relatively high titratable acidity; specific values would greatly help differentiate seasonal effects.
This version is improved, has a lot of data but somewhat weak on discussion. There is more than enough to discuss just on the flavonoid identification without the need for ABTS and FRAP data.
Author Response
Dear Reviewer,
First and foremost, we would like to express our sincere gratitude for the thoughtful and detailed comments, as well as constructive criticism, which have greatly contributed to improving the manuscript. Below, we provide our responses to the comments.
(Note: Reviewer comments are in italics; authors' responses are in regular text.)
The manuscript has been updated with all the corrections, followed by the accepted changes to enhance readability.
Round2
Reviewer 1
Authors have made numerous changes and improvements to the ms. There are some additional changes that need to be made.
Line 63, 2012 is not exactly recent. There do not seem to be clear statistics on the percent of the raspberry industry used for fresh markets, although it appears that the 15% global raspberry production from Mexico is 90% fresh market and most of US production is also fresh (see https://www.usitc.gov/publications/332/pub5194.pdf). Not sure if the no. 2 reference gave a guesstimate of fresh market raspberry; I would say 20% of global production is fresh market but most likely centered in the Americas.
-Thank you for your suggestion. This paragraph has been revised, and the references have been updated with newer and more accurate ones.
Line 65, go back to Tan and refer to the FAO reference they used. Currently the sentence used here is close to word to word from Tan 2022. Line 66, Mexico seems to be second to Russia at the moment, looking at other sources. It might be better to state the top 5 world leaders in red raspberry production are Russa, Serbia, Mexico, United States, Poland.
-Thank you for your suggestion. This paragraph has been revised. According to FAO 2023 data, Russia was the world’s leading producer of raspberries, followed by Mexico, Serbia, Poland and the United States, which together ranked as the top five producers globally.
Line 71, replace ‘The varieties’ with ‘genotypes’.
-Thank you for the suggestion. It has been replaced.
Line 55 and line 80 are almost the same. Suggest changing line 80.
-Thank you for the suggestion. Line 80 has been changed.
Line 84 research does not identify; those doing the research do this.
-Thank you for the suggestion. The sentence has been rewritten.
Lines 93-98. R. chingii is different from red raspberry, apparently because carotenoids increase with ripening and provide most of the red color. This addition confuses the point of the present experiment. Anthocyanins in most berries increase, but not all flavonoids increase with ripening. Here, the point is to utilize newer technologies to follow seasonal effects on flavonoid profiles of a newer primocane variety.
-Thank you for your suggestion. We have removed the citation for R. chingii from this paragraph to prevent any confusion
Line 120, it’s a pretty big stretch to move to metabolites-postharvest, for a couple of reasons. One is that different temperatures during production and harvest (field heat) can mitigate specific metabolic profiles. This sentence (L 472-473) in the conclusion needs to be removed.
-Thank you for your suggestion. We have removed the sentence from the conclusions.
Methods: were internal standards used?
-Thank you for your suggestion. No internal standard was used during sample preparation; however, the Agilent tune mix was employed for instrument calibration. A reference solution (m/z 121.0509 and m/z 922.0098 for positive mode, and m/z 119.036 and m/z 980.0163 for negative mode) was applied to correct minor mass drifts during data acquisition, ensuring high-precision measurements. This information was already indicated in the “Instrumentation and Experimental Conditions” in M&M section.
Line 180 implies several studies showed higher flavonoids in later harvests-references? Line 194, why would tyrosols and hydroxycoumarins be higher in the RiJ samples?
-Thank you for your suggestions. The paragraph has been revised. Additionally, the fact that hydroxybenzaldehydes, hydroxycoumarins and tyrosols showed significantly greater abundance in the RiJ may be related to a defence mechanism against diferente biotic or abiotic stresses that could be more prevalent during that harvest period, although this remains a hypothesis. The role of these compunds is explained in page 10 (lines 233-242), which is why it is not discussed in this paragraph.
Figure 4-need to add in A, B and increase font on y axis, also enlarge legend.
-Thank you for the suggestion. The Figure has been revised according to the reviewer´s comments.
Fig 5, still can’t make out the y axis names. Seems to be partly a font and partly an uneven black color.
-Thank you for the suggestion. Figure 5 has been revised.
Lines 224-243 need to be better singulated between plant pathogens and human bioactives.
-Thank you for your valuable suggestion. We agree with the reviewer, and the paragrapgh has been revised for clarity.
Line 292 should start a new paragraph. Lines 298-304 are also hard to follow-a lot of ‘this’ which makes it difficult to follow what is alluded to, and wandering sentences. Different plateaus-what are these?
-Thank you for your suggestions. The paragraph has been revised according to the reviewer´s comments. In the context of the cited article, "raspberries from different plateaus" refers to raspberries cultivated in two distinct elevated regions, suggesting that the geographical and environmental conditions of these areas may influence the antioxidant characteristics and activities of the raspberries.
Line 337-maturity index is not helpful here. Raspberries are 8-12% ssc and 0.8-1.2% titratable acidity. These ratios of 4 and 3 indicate relatively high titratable acidity; specific values would greatly help differentiate seasonal effects.
-Thank you for your suggestion. The specific values of SSC and TA for each harvest period are proved in the Plant Materials (M&M section). As can be seen, the differences between the two harvest periods are primarily related to a slightly higher sugar content in the September fruits. These fruits also content a slightly higher amount of acids, but the difference in MI appears to be more influenced by the increase in sugar.
This version is improved, has a lot of data but somewhat weak on discussion. There is more than enough to discuss just on the flavonoid identification without the need for ABTS and FRAP data.
-Thank you for the suggestion. It was important for us to also include the relationship between the presence of certain polyphenols and the antioxidant capacity of the fruits. This is crucial as it serves as a defense mechanism against stresses. Additionally, antioxidant properties are a key trait in raspberry fruits that influence their appeal and, consequently, the increase in their consumption

Reviewer 2 Report
Comments and Suggestions for Authors
The author has made moderate explanations and some supplements to the questions I raised before, which can basically solve my questions.
Author Response
Thank you very much for your valuable feedback. We appreciate your recognition of the clarifications and additional explanations provided.
Reviewer 3 Report
Comments and Suggestions for Authors
The authors have adequately addressed the comments made in the previous review. The comments made have been explained or addressed in the text to a satisfactory degree.
Author Response
Thank you for your insightful feedback. We are grateful for your acknowledgment of the clarifications and additional explanations we provided.